# Spiramycin Disarms *Pseudomonas aeruginosa* without Inhibiting Growth

**DOI:** 10.3390/antibiotics12030499

**Published:** 2023-03-02

**Authors:** Matteo Calcagnile, Inès Jeguirim, Salvatore Maurizio Tredici, Fabrizio Damiano, Pietro Alifano

**Affiliations:** 1Department of Biological and Environmental Sciences and Technologies, University of Salento, Via Monteroni, 73100 Lecce, Italy; 2École de Biologie Industrielle, 95895 Cergy, France

**Keywords:** *Pseudomonas aeruginosa*, spiramycin, macrolide antibiotics, anti-virulence drugs, *Galleria mellonella*

## Abstract

Spiramycin is a 16-membered macrolide antibiotic currently used in therapy to treat infections caused by Gram-positive bacteria responsible for respiratory tract infections, and it is also effective against some Gram-negative bacteria and against *Toxoplasma* spp. In contrast, *Pseudomonas aeruginosa*, which is one of the pathogens of most concern globally, is intrinsically resistant to spiramycin. In this study we show that spiramycin inhibits the expression of virulence determinants in *P. aeruginosa* in the absence of any significant effect on bacterial multiplication. In vitro experiments demonstrated that production of pyoverdine and pyocyanin by an environmental strain of *P. aeruginosa* was markedly reduced in the presence of spiramycin, as were biofilm formation, swarming motility, and rhamnolipid production. Moreover, treatment of *P. aeruginosa* with spiramycin sensitized the bacterium to H_2_O_2_ exposure. The ability of spiramycin to dampen the virulence of the *P. aeruginosa* strain was confirmed in a *Galleria mellonella* animal model. The results demonstrated that when *G. mellonella* larvae were infected with *P. aeruginosa,* the mortality after 24 h was >90%. In contrast, when the spiramycin was injected together with the bacterium, the mortality dropped to about 50%. Furthermore, marked reduction in transcript levels of the antimicrobial peptides gallerimycin, gloverin and moricin, and lysozyme was found in *G. mellonella* larvae infected with *P. aeruginosa* and treated with spiramycin, compared to the larvae infected without spiramycin treatment suggesting an immunomodulatory activity of spiramycin. These results lay the foundation for clinical studies to investigate the possibility of using the spiramycin as an anti-virulence and anti-inflammatory drug for a more effective treatment of *P. aeruginosa* infections, in combination with other antibiotics.

## 1. Introduction

Drug repositioning is the practice of finding new uses for existing drugs [1]. The benefit of drug repositioning is to drastically reduce the cost of research and development as well as the high risk of failure that is commonly associated with new drug discovery, and, mostly, to reduce the cost of preclinical trials to ensure their safe use. In addition to existing drugs, drug repositioning allows the possibility to reuse helved drugs, or drug candidates that failed clinical trials for other medical indications. Drug repositioning is particularly valuable in the fight against pathogenic bacteria, given the high cost and considerable time associated with the discovery and introduction into clinical practice of new antibiotics [2,3,4,5]. Furthermore, drug repositioning is helpful in the challenge against the global emergency caused by the rapid spread of multidrug-resistant bacteria despite a decrease in the number of approved antibiotics [2,3,4,5].

Both non-antibiotic and antibiotic drugs have been recently repositioned for new uses against pathogenic bacteria. Examples of non-antibiotic drug repositioning include some psychotropics, local anesthetics, tranquilizers, cardiovascular drugs, antihistamines, and anti-inflammatories whose antibacterial activity has also been proven [2,3,4,5]. In this context, it may be of interest to mention the Food and Drug Administration-approved iron mimetic metal Gallium (Ga(III)) that has been successfully repurposed as an antimicrobial drug. First-, second-, and third-generation of Ga(III) formulations were proven to be effective against multidrug-resistant ESKAPE pathogens (ESKAPE is an acronym that includes six nosocomial bacterial pathogens: *Enterococcus faecium*, *Staphylococcus aureus*, *Klebsiella pneumoniae*, *Acinetobacter baumannii, Pseudomonas aeruginosa*, and *Enterobacter* spp.) [6].

Examples of antibiotic drug repositioning include some macrolides whose activity as anti-virulence drugs has been demonstrated in macrolide-resistant bacteria. In particular, *Pseudomonas aeruginosa*, which is one of the most common pathogens in chronic lung infection and the third most common pathogen associated with nosocomial urinary tract infections (UTIs), is considered intrinsically resistant to macrolides [7]. Indeed, most of *P. aeruginosa* strains demonstrate minimal inhibitory concentrations (MIC) in the range 128–512 mg/L due to the acquirement of specific mutations in the 23S rRNA gene, rendering them very poorly susceptible to macrolides [8]. Despite the results of in vitro susceptibility tests, interest in macrolides in the treatment of some pseudomonal infections arose from the observation, in the early 1980s, that patients with diffuse panbronchiolitis improved during long-term treatment with macrolides [9]. Moreover, in a mouse model of *P. aeruginosa* bacteremia, treatment with erythromycin led to a survival rate of 80% compared with 20% survival rate in the control [10]. These findings led to the speculation that macrolides may act during pseudomonadal infections by down-modulating either the inflammatory response or *P. aeruginosa* virulence [11].

Macrolides have the property to accumulate within cells of the immune system and have complex immunomodulatory activities [11,12,13,14]. Moreover, the 14-membered macrolide erythromycin and its derivative azithromycin have the ability to inhibit, at clinically relevant concentrations that are not able to affect bacterial growth, the expression of many virulence factors of *P. aeruginosa.* These factors include exotoxin A, protease, elastase, lipase, phospholipase C, lecithinase, gelatinase, DNase, and pyocyanin (PYO) [11,15,16,17,18,19,20].

Some explanations have been provided, over the last years, about the inhibitory effect on the expression of *P. aeruginosa* exoenzymes by macrolides. Some of them invoked possible inhibitory effects on protein synthesis in the case of short-chain peptides [21], others invoke some inhibitory effects on the activity of specific enzymes. For instance, 14- and 15-membered macrolides (but not 16-membered macrolides) were reported to inhibit specifically the guanosine diphosphomannose dehydrogenase that is required for biosynthesis of alginate [9]. More recently, the anti-virulence activity of erythromycin and azithromycin has been attributed to their ability to inhibit the quorum sensing (QS) circuit in *P. aeruginosa* [22,23,24,25]. This finding was confirmed in a cystic fibrosis model of chronic *P. aeruginosa* lung infection in which treatment with azithromycin resulted in suppression of quorum sensing (QS)-regulated virulence factors, impairment to form mature alginate biofilm, increased sensitivity to complement, and stationary-phase killing [26]. These results have promoted the use of these macrolides in adjunct therapy against chronic and/or biofilm-mediated *P. aeruginosa* infections [27,28,29,30,31,32,33]. However, some concerns have been raised that the use of these drugs could select for more virulent strains in the nosocomial environment [34].

While most of the studies on the anti-virulence activity of macrolides focus on erythromycin and on its derivative azithromycin, there is almost no information on spiramycin, which differs from erythromycin by having a larger macrolactone ring and a different sugar decoration. Spiramycin is a 16-membered macrolide used in human medicine as an antibacterial and antiparasitic agent (active against *Toxoplasma* spp.) [35,36,37]. The antibacterial activity of this antibiotic belonging to the macrolide–lincosamide–streptogramine B class (MLS_B_) was associated with its ability to bind the 50S ribosomal subunit and to block the path by which nascent peptides exit the ribosome [38]. Similar to josamycin and clindamycin, spiramycin causes dissociation of peptidyl-tRNAs containing two, three, or four amino acid residues [38].

Spiramycin is mostly effective against Gram-positive bacterial pathogens responsible for respiratory tract infections including *Staphylococcus aureus*, streptococci of groups A, B, C, and D, and pneumococcus [39]. It is also effective against bacteria belonging to the genera *Neisseria*, *Legionella*, *Mycoplasma*, *Chlamydia*, and against *Toxoplasma* spp. [39]. In contrast, *Pseudomonas aeruginosa* is considered intrinsically resistant to spiramycin [7]. In this study, after demonstrating that spiramycin completely inhibits the production of PYO and pyoverdine (PVD) in *P. aeruginosa* without affecting growth in vitro, we tested its ability to inhibit *P. aeruginosa* virulence in a *Galleria mellonella* animal model.

## 2. Results

### 2.1. Spiramycin Inhibits the Production of Pyocyanin and Pyoverdine in P. aeruginosa

Minimum inhibitory concentration (MIC) experiments showed that the *P. aeruginosa* strain GG-7R was moderately or poorly sensitive to several antibiotics but highly resistant to spiramycin [40] (Table 1). MIC values for ampicillin, streptomycin, and rifamycin were 500 μg/mL, 62.5 μg/mL, and 31.3 μg/mL, respectively. Bacterial growth was also inhibited by high concentrations of erythromycin (MIC value 250 μg/mL), but never by even higher concentrations of spiramycin (500 μg/mL) (Table 1), even though these antibiotics belong to the same class, the macrolides.

Despite the inability to inhibit growth, spiramycin markedly affected PYO and PVD pigment production by *P. aeruginosa* GG-7R (Figure 1). The multi-well plates from the MIC experiments were used to measure PYO, PVD, and biomass after growth for 24 h in Luria–Bertani (LB) broth. The results confirmed that spiramycin did not inhibit the bacterial growth in the concentration range of 1 to 500 μg/mL, even though it did slightly reduce the final biomass values measured as absorbance at 600 nm (Figure 1A). To study the pigment production, the absorption and fluorescence emission spectra of the exhausted broth of *P. aeruginosa* GG-7R were first analyzed (Figure 2). The absorption spectrum was analyzed in a wavelength range from 295 to 800 nm (Figure 2A). The results showed that the absorption spectrum of the exhausted broth of *P. aeruginosa* GG-7R grown in the absence of spiramycin was similar to that found in other *P. aeruginosa* strains [41]. The absorption between 295 and 421 nm and between 548 and 800 nm was markedly reduced when the bacterium was cultured with spiramycin (30 μg/mL). The fluorescence emission spectrum was then analyzed by using 405 nm as the excitation wavelength. The results demonstrated an inhibition of the production of fluorescent pigment in the presence of spiramycin (30 μg/mL) (Figure 2B). Quantitative estimates showed that PVD production decreased dramatically when the spiramycin concentration reached and exceeded 15.6 µg/mL (Figure 1B), and PYO production also decreased with spiramycin concentration of 7.8 μg/mL or higher (Figure 1C).

The phenotype of bacteria growing at sub-MIC concentrations of spiramycin or erythromycin was evaluated on LB agar plates (Figure 3A,B). After 24 h of growth, spiramycin did not affect bacterial growth at all tested concentrations (30, 60, 90, 120, 150, and 180 μg/mL). However, production of greenish pigmentation was markedly reduced (Figure 3A). The ability of spiramycin (120, 150 and 180 μg/mL) to inhibit pigment production without inhibiting bacterial growth was even more evident after 48 h of incubation (Figure 3B). After 48 h (72 h, 96 h, and 120 h) no changes were detectable in the plate with spiramycin at 120, 150 and 180 µg/mL. However, this effect could also be related to the fact that the bacterial culture is in the stationary phase after 24 h. Production of greenish pigmentation was also markedly reduced in erythromycin-exposed bacteria after 24 h of incubation. However, bacterial growth was clearly inhibited at erythromycin concentrations of 30, 60, and 90 μg/mL, and markedly inhibited at erythromycin concentration of 120 μg/mL or higher (Figure 3C).

Biomass (absorbance at 600 nm), pH, and PYO production were then evaluated during *P. aeruginosa* GG-7R growth in the presence of two concentrations of spiramycin (30 and 120 ug/mL) in LB broth in 250 mL Erlenmeyer flasks. The results confirmed that spiramycin did not inhibit the bacterial growth, even though it did slightly reduce the final biomass values (Figure 4A). pH also did not vary in spiramycin-treated cultures compared to untreated cultures (Figure 4B), while PYO production was markedly reduced (Figure 4C). Overall, these results demonstrated that spiramycin treatment inhibits PYO and PVD production by *P. aeruginosa* without substantially inhibiting growth in vitro.

After growth in the flask, RT-qPCR was performed on two genes related to PYO and rhamnolipid synthesis. The *phzS* gene, which catalyzes the last step of PYO synthesis [42], and the *rhlC* gene involved in di-rhamnolipid formation were analyzed [43,44]. To further extend the results, *lasB* was added to the RT-qPCR analysis. LasB is an elastase involved in biofilm formation, activates interleukin-1β (IL-1β), and cleaves host molecules such as elastase and collagen [45,46,47]. LasB is also a new target for the development of new therapeutic strategies [48,49]. The RT-qPCR results demonstrated that the transcript levels of the *rhlC*, *phzS*, and *lasB* genes were markedly down-regulated at 24 h and 48 h (Figure 4E,F).

### 2.2. Spiramycin Sensitizes P. aeruginosa to Oxidative Damage

There is evidence that PYO is an active redox compound that protects *P. aeruginosa* against photodynamically induced oxidative damage [50]. Since we found that spiramycin inhibits PYO production, we wanted to analyze the resistance to hydrogen peroxide exposure of *P. aeruginosa* GG-7R grown either in the presence or in the absence of spiramycin. For this purpose, bacterial cultures containing an increasing concentration of H_2_O_2_ (10, 20, 30, and 40 mM) were prepared, and spiramycin was added to these cultures at three different concentrations (60, 120, and 180 μg/mL). The biomass increase was measured by absorbance at 600 nm after 24 h of incubation. The results are shown in Figure 5. When cultures were exposed to 10, 20, and 30 mM H_2_O_2_, the biomass at 24 h decreased significantly as a function of spiramycin concentration and was the highest when the cultures were untreated with spiramycin. As expected, in samples untreated with spiramycin the biomass at 24 h decreased as a function of H_2_O_2_ concentration. At 40 mM H_2_O_2_, the effect of oxidative damage became predominant, and final biomass values were low in all samples either with or without spiramycin treatment. These results demonstrated that spiramycin sensitizes *P. aeruginosa* to oxidative damage, and this could be due to the decreased production of active redox compounds such as PYO.

### 2.3. Spiramycin Inhibits the Formation of P. aeruginosa Biofilm on Hydroxyapatite

Erythromycin and azithromycin have been shown to inhibit or delay biofilm formation by *P. aeruginosa* [51,52,53]. To verify that spiramycin also had this effect, bacterial cultures were set up using hydroxyapatite as a biomimetic substrate for biofilm formation. The bacterial cultures were then incubated either in the absence or in the presence of spiramycin (30 μg/mL) for 72 h at 37 °C and 150 rpm before subsequent analyses. After the incubation, the bacterial cultures incubated in the presence of spiramycin did not produce the intense green pigmentation that was produced by the bacterial cultures grown in the absence of spiramycin (Figure 6A). Quantitative estimates showed that both PYO and PVD production were strongly inhibited by the presence of spiramycin (Figure 6B,C). Biofilm formation and bacterial growth on hydroxyapatite was then determined by using the crystal violet staining method and the CFU count, respectively. The results showed that both biofilm formation and bacterial growth on hydroxyapatite were inhibited by exposure to spiramycin (Figure 6D,E). In contrast, the biomass of planktonic bacteria determined by absorbance at 600 nm was higher in the presence than in the absence of spiramycin (Figure 6F). These results suggested that spiramycin exposure disfavored biofilm formation in favor of planktonic growth.

### 2.4. Spiramycin Inhibits the Swarming Motility and the Biosynthesis of Rhamnolipids of P. aeruginosa

To find out whether spiramycin also affected the motility of *P. aeruginosa*, the swarming motility was evaluated on agarized medium as described [46]. The results demonstrated that spiramycin severely inhibited the swarming motility (Figure 7A,B). Rhamnolipids are essential for biofilm architecture and mediate the dispersal of bacteria when the biofilm is mature [47,48,49,50,51,52,53,54,55]. Furthermore, rhamnolipid production promotes twitching/swarming motility [55,56,57,58]. Thus, rhamnolipids were quantified in the exhausted broths of *P. aeruginosa* GG-7R grown either in the absence or in the presence of spiramycin (60 μg/mL). The assay was carried out after 24, 48, and 72 h of growth in LB broth in 250 mL Erlenmeyer flasks. The results indicated that spiramycin markedly reduced the production of rhamnolipids (Figure 7C). This result was also confirmed with an RT-qPCR that analyzed the transcript levels of the *rhlC* gene involved in the di-rhamnolipid synthesis (Figure 4E,F).

### 2.5. Spiramycin Attenuates P. aeruginosa Virulence and Expression of Antimibrobial Peptides in Galleria mellonella Animal Model

The above results showed that spiramycin inhibits the production of PYO, PVD, and rhamnolipid. Furthermore, this macrolide also inhibited biofilm formation. Since all these factors play an important role in the pathogenicity and virulence of *P. aeruginosa* [59], two experiments were designed using *G. mellonella* larvae as an in vivo infection model. Indeed, *G. mellonella* larvae are a widely used model in microbiological research to assess bacterial virulence and innate immunity since they have an innate immune system that is similar to that of vertebrates [60].

Thus, *P. aeruginosa* GG-7R was used to infect *G. mellonella* larvae either in the absence or in the presence of spiramycin. In a first experiment, *G. mellonella* larvae were infected with an inoculum suspension containing 80 CFU/larva. Mortality was determined after incubating the larvae for 24 h at 37 °C. The results showed that the mortality of infected larvae without the addition of spiramycin was about 90%, while the addition of spiramycin reduced this value to 50% (Figure 8A). Implemented controls showed that the non-injected larvae (control larvae), the larvae inoculated with injection solution alone (mock inoculum), and the larvae injected with spiramycin alone (toxicity control) showed comparable mortality (about 10%) (Figure 8A).

In a second experiment, *G. mellonella* larvae were infected with a higher concentration of *P. aeruginosa* (8 × 10^3^ CFU/larva) and incubated for 6 h at 37 °C. After this time, the larvae were immediately frozen at −80 °C and the RNA was extracted. Then, the mRNA levels of genes encoding three antimicrobial peptides (gallerimycin, gloverin, moricin) and lysozyme were determined by RT-qPCR. Three groups of larvae were used in this experiment: larvae inoculated with injection solution alone (mock inoculum), larvae infected with the bacterium, and larvae infected with the bacterium in the presence of spiramycin (60 μg/mL). The results demonstrated increased transcript levels of all genes analyzed in the larvae infected with *P. aeruginosa* compared to uninfected larvae (Figure 8B). In contrast, the simultaneous injection of the bacterium and spiramycin reduced the transcript levels of these genes compared to the bacterium without spiramycin (Figure 8B). These results indicated that spiramycin was able to attenuate the virulence of *P. aeruginosa* and to down-modulate the expression of determinants of innate immunity in the animal model of *G. mellonella*.

## 3. Discussion and Conclusions

*P. aeruginosa* is one of the most concerning pathogens globally. In fact, it is on the WHO Priority 1 list, mainly because some strains are resistant to carbapenems. Innovative strategies to combat *P. aeruginosa* include bacteriophages, nanoparticles, QS inhibitors, and antimicrobial peptides [61,62,63,64]. Some of these strategies, such as the use of QS inhibitors, have the advantage of inhibiting virulence, while other molecules, such as antimicrobial peptides, show a low level of induced resistance [64]. Using molecules that do not kill bacteria but reduce their pathogenicity could be advantageous because these molecules eliminate the possibility of selective pressure that would favor the emergence of antibiotic-resistant strains.

This study investigates anti-virulence properties of spiramycin on *P. aeruginosa*. The results show that spiramycin negatively affects PVD and PYO production by *P. aeruginosa* (Figure 1, Figure 3, Figure 4, and Figure 6). These molecules are important virulence factors: PYO is a toxin and PVD is a siderophore [59,65]. Furthermore, PYO and PVD play an endogenous role in the physiology of the bacterium. In fact, it has been shown that the reduction PYO production sensitizes the bacterium to oxidative damage [50]. Indeed, we found that treatment of *P. aeruginosa* with spiramycin sensitizes the bacterium to H_2_O_2_ exposure (Figure 5). This finding is relevant because the antibiotics, regardless of their specific mechanism of actions, all trigger oxidative damage [66,67]. Consequently, spiramycin might also sensitize *P. aeruginosa* to the exposure to other antibiotics. In addition to PVD, PYO, and other virulence factors, some bacterial mechanisms also favor the onset of *P. aeruginosa* infection. Among these aspects, there is the ability of the bacterium to form biofilm and to diffuse by swarming motility thanks to the flagella and the production of biosurfactants [51]. Notably, spiramycin reduces biofilm formation, motility, and rhamnolipid production in *P. aeruginosa* (Figure 7). Spiramycin also markedly reduces the transcript levels of the elastase *lasB* [45,46,49].

Although the abilities to decrease PVD, PYO, rhamnolipid, *lasB* transcript, and biofilm production as well as to inhibit swarming motility are well known for the 14-membered macrolides erythromycin and its derivative azithromycin [11,17,68], our results revealed these abilities, for the first time, in a 16-membered macrolide, spiramycin. Furthermore, our results highlight that the use of spiramycin offers advantages over that of erythromycin and azithromycin in the study of the anti-virulence properties of macrolides in *P. aeruginosa*. Indeed, the growth of *P. aeruginosa* was not affected even with the highest concentration of spiramycin tested (500 μg/mL), while expression of virulence determinants was markedly reduced (Figure 1, Figure 3 and Figure 4). In contrast, growth of *P. aeruginosa* GG-7R was clearly inhibited at erythromycin concentrations of 30, 60, and 90 μg/mL, and markedly inhibited at erythromycin concentration of 120 μg/mL or higher (Figure 3). Thus, spiramycin offers the advantage over erythromycin and azithromycin of studying the anti-virulence properties of macrolides and, in particular, their inhibitory effect on bacterial QS under conditions in which bacterial growth remains substantially unchanged.

Indeed, there is growing evidence that the anti-virulence activity of macrolides could be due to their ability to inhibit the QS circuit in *P. aeruginosa* [22,23,24]. However, the precise mechanism underlying the inhibitory effect of macrolides on *P. aeruginosa* QS remains to be defined. The inhibitory effects of azithromycin on QS have been associated with inhibition of the mRNA expression of N-acyl homoserine lactone synthesis enzymes, upstream of *lasI* or *rhlI* [69], decreased expression of QS regulatory genes *lasI*, *lasR*, *rhlI,* and *rhlR* [70], dysfunctional and/or impaired secretion of acyl homoserine lactones (AHL) 3-oxo-C12-homoserine lactone (3-oxo-C12-HSL), and C4-homoserine lactone (C4- HSL) [71], changes in membrane permeability, thereby influencing the flux of 3-oxo-C12-HSL [72] and inhibition of expression of the GacA-dependent small RNAs RsmY and RsmZ, both of which acts upstream of the quorum-sensing machinery [73].

Interestingly, selective inhibition of translation of subsets of mRNAs depending on their codon usage [74], including *rhlR* transcript [54,75], has been proposed more recently. According to this hypothesis, the inhibitory effect of macrolides at sub-MIC concentration on QS and on the expression of virulence determinants would be due to their mechanism of inhibition of translation, albeit selective on mRNA subsets, and not due to an off-target mechanism. On the other hand, screening of FDA-approved drugs led to identify nitrofurazone and erythromycin estolate as direct PqsE inhibitors [76]. PqsE is involved in biosynthesis of the QS signal molecules 2-alkyl-4(1*H*)-quinolones (AQs) of the PQS system, but it also increases the expression of virulence determinants and biofilm genes [77]. In addition, PqsE exerts a repression on *pqsA* promoter activity that is, in contrast, stimulated by PqsR [77]. Our evidence that spiramycin markedly reduced the expression of virulence determinants in the absence of any significant effect on bacterial multiplication could support an off-target mechanism, although both mechanisms (ribosomal and off-target) can coexist.

In this study, the lepidopteran *G. mellonella* was used as an in vivo infection model to investigate the anti-virulence properties of spiramycin. *G. mellonella* has an innate immune system that shares essential properties with that of vertebrates [52]. Thus, this insect has been used as a suitable infection model with various pathogens, including *P. aeruginosa* [78,79]. In particular, when exposed to different pathogens, *G. mellonella* larvae produces several antimicrobial peptides, whose transcript levels can be measured by qPCR [79,80,81]. Our results show that when *G. mellonella* larvae were infected with *P. aeruginosa* the mortality after 124 h was >90%. In contrast, when the spiramycin was injected together with the bacterium, the mortality dropped to about 50% (Figure 8A). This result confirmed the anti-virulence properties of spiramycin, which may be due to its ability to inhibit the biofilm formation and the expression and virulence determinants, as observed in in vitro experiments.

It is also interesting to note a marked reduction in transcript levels of three antimicrobial peptides (gallerimycin, gloverin, and moricin) and lysozyme in *G. mellonella* larvae infected with *P. aeruginosa* with spiramycin, compared to larvae infected without the addition of spiramycin (Figure 8B). This result can be interpreted either in light of the ability of spiramycin to dampen the expression of virulence determinant and biofilm formation as above discussed, or in light of the anti-inflammatory properties of macrolides on host cells. Indeed, there is evidence that macrolides accumulate within cells of the immune system where they perform immunomodulatory activities [11,12,13,14,82]. Thus, the use of spiramycin in the *G. mellonella* may also be a suitable system to investigate the immunomodulatory activities of the macrolides in an in vivo model.

Since the macrolides azithromycin and erythromycin are active on other strains of *P. aeruginosa,* it is reasonable to assume that the results concerning spiramycin can be extended to different strains of *P. aeruginosa.* For example, azithromycin and erythromycin are active against *P. aeruginosa* PAO1 [83,84,85,86,87] and against various clinical isolates [85,88]. In addition, spiramycin forms non-host-guest complexes with methyl-β-cyclodextrin; therefore, these complexes could be used as antibiotic delivery systems as already proposed in the case of other macrolides [89,90,91].

In conclusion, the in vitro and in vivo results provided in this study, in addition to contributing to our understanding of the mechanism of action of macrolides, lay the foundation for clinical studies to investigate the possibility of using the spiramycin as an anti-virulence and anti-inflammatory drug for a more effective treatment of *P. aeruginosa* infections, in combination with other antibiotics.

## 4. Materials and Methods

### 4.1. Strain, Media, Growth Condition and General Procedure

The environmental strain *P. aeruginosa* GG-7R was used in this study [92]. Partial sequence of the 16S ribosomal RNA gene is deposited in the GenBank under the accession number MF045142.1. The bacterium was grown in Luria–Bertani (LB) medium (10 g NaCl (Oxoid^TM^, Altrincham, England), 10 g Tryptone (BD Difco™ Bacto™, Franklin Lakes, NJ, USA), 5 g Yeast Extract (BD Difco™ Bacto™), 15 g Agar (BD Difco™ Bacto™) when required, distilled H_2_O up to 1 L) at 37 °C and 200 rpm. Media sterilization was performed in an autoclave (121 °C, 20 min). Stock solutions of spiramycin (Sigma-Aldrich, St. Louis, MO, USA) (100 mg/mL) and erythromycin (Sigma-Aldrich) (125 mg/mL) were prepared by solubilizing the antibiotics with ethanol (96%). Sterilization of the antibiotic solutions was carried out by filtration (0.22 μm, sterile syringe filter, WWR^®^).

LB medium, solid or liquid, with erythromycin or spiramycin, was prepared as described above and then allowed to cool for 30 min at 45 °C. A volume of antibiotic solution was added to the medium to obtain the final concentration. As a control, the same volume of solvent (96% ethanol) was added to the antibiotic-free LB medium. After adding the antibiotic solution or solvent, the medium was stirred for 15 min using a magnetic stirrer at 120 rpm. The solid medium was poured into 90 mm Petri dishes, while the liquid medium was allowed to cool at 37 °C before use.

The inoculum was prepared for all experiments as described: (i) the bacterium was seeded onto solid LB; (ii) the following day, a colony was taken using a sterile toothpick and inoculated into 10 mL of liquid LB; (iii) the next day, the bacterial suspension was inoculated (1 in 100) into an appropriate volume of liquid LB; (iv) the bacterium was grown until the absorbance at 600 nm was 0.3 ± 0.05. The resulting suspension served as inoculum for the different growth experiments. All growth experiments were performed in triplicate.

### 4.2. Determination of MIC, and Measurement of PVD and PYO Production in Multiwell Plates

The MIC of spiramycin was determined by using 24-well plates and LB broth with 10 different concentrations of spiramycin (from 0 to 500 μg/mL). MIC plates were incubated at 37 °C and 180 rpm for 24 h before analyzing the results. The amount of PVD in the exhausted broths was measured using the fluorescence readout (λexc = 405 nm, λem = 613 nm) [93]. The amount of PYO in the exhausted broths was measured using the absorbance measurement (520 nm) [54]. A turbidity measurement (absorbance at 600 nm) was performed to measure biomass increase. All the measures were performed using the Cytation5 multimodal and imaging reader (BioTek, Winooski, VT, USA). Absorption at 520 nm (PYO) and fluorescence (PVD) values were reported as biomass (absorbance at 600 nm) normalized value.

Mic experiments for the other antibiotics (ampicillin, streptomycin, erythromycin, and rifamycin) were carried out following the same protocol and measuring absorbance at 600 nm.

### 4.3. Determination of Growth Curves and Measurement of PVD, PYO and Rhamnolipid Production in Shake Flask Experiments

*P. aeruginosa* was grown in 250 mL flasks containing 50 mL LB. The flasks were incubated at 37 °C and 180 rpm for 72 h. Samples were taken at the set times. Biomass increase (absorbance at 600 nm), pH, and PYO and PVD production was measured at different time points. Broth culture samples were centrifuged at 4000 rpm at 4 °C for 20 min, and supernatants were used to quantify PYO and PVD production. PYO was quantified by measuring the absorbance at 520 nm (V-10 PLUS spectrophotometer, ONDA). PVD was quantified by measuring the fluorescence (λexc 405 nm, λem 613) (JASCO Inc., Easton, MD, USA, FP-750 Spectrofluorometer). The continuous absorption spectrum was measured using a spectrophotometer (Beckman Coulter DU^®^ 800 spectrophotometer) in the range from 295 nm to 800 nm. The fluorescence emission spectrum was measured using an excitation wavelength (λexc) of 405 nm, and the emission spectrum (λem) was recorded in a range from 450 to 700 nm using a spectrofluorometer (JASCO Inc., FP-750 Spectrofluorometer). Before measuring the absorption and fluorescence spectrum, the exhausted LB broth was centrifuged at 4000 rpm for 15 min and the supernatant was diluted 1:200 using sterile water. The amount of PYO and PVD was normalized by the biomass (absorbance at 600 nm). The LB medium spectrum alone was used as a blank to detect continuous spectra of both absorption and fluorescence. The orcinol method was used to determine the amount of rhamnolipids [54]. To quantify the rhamnolipids, a calibration curve was set up by preparing serial dilutions of L-rhamnose concentrations ranging from 0 to 100 μg/mL as described [54].

### 4.4. Resistance to Oxidative Stress

The resistance to oxidative stress of *P. aeruginosa* was tested in 96-multiwell plates. Specifically, the inoculum was prepared as described above and diluted 1:50 before starting the experiment. A 96-well plate was used for this purpose, each well containing 100 μL of the bacterial suspension. H_2_O_2_ was added to some wells at four different concentrations (10, 20, 30, and 40 mM). Spiramycin was added to wells at three concentrations (60, 120, and 180 μg/mL). The broth cultures thus prepared were incubated at 37 °C and 180 rpm for 24 h. After this time, the cultures were sampled, and the biomass was measured by determination of the absorbance at 600 nm.

### 4.5. Biofilm Experiment and Biomass Estimation

Microcrystalline hydroxyapatite discs (diameters = 8 mm, Bonding chemical) were used as biomimetic support to perform the biofilm experiments. The hydroxyapatite discs were sterilized with the aid of the autoclave (120 °C, 20 min) and placed inside 24-well plates. The bacteria were cultivated as described above, and 1 mL of the bacterial suspension was used to inoculate 25 mL of LB broth. Next, 1 mL of the resulting suspension was poured into each well containing the hydroxyapatite disc. The 24-well plates were incubated at 37 °C and 150 rpm for 72 h before the analysis.

Two methods were used to assess biofilm formation: the crystal violet method [94] and the CFU method (CFU per disc). To measure biofilm using the crystal violet protocol (CV), discs were washed three times by sterile saline solution (0.9% NaCl) and treated with methanol for 15 min to fix the bacteria. Subsequently, the discs were immersed into a solution of crystal violet (Liofilchem) and ethanol (1:5). This solution was left to act for 5 min. The discs were washed three times by the sterile saline solution and immersed in 100% ethanol for 10 min. At the end of this process, the ethanol solution was used to measure the absorbance at 595 nm (V-10 PLUS, ONDA spectrophotometer). For the CFU method, the discs were washed three times by the sterile saline solution and immersed in 1 mL of LB. At this point, the discs were vortexed at maximum speed for 2 min to promote the detachment of bacteria from the discs. The resulting solution was used to make serial dilutions using LB as the diluent. Finally, the dilutions were plated onto LB agar. The resulting LB plates were incubated at 37 °C, and CFU were counted after 24 h of incubation. Quantitative evaluation of PYO and PVD production was carried out as described above. The amount of PYO and PVD was normalized by the biomass measured with the CV protocol.

### 4.6. Swarming Motility Evaluation

Motility experiments were performed using the BM2 solid medium [54]. The medium was composed as follows: 62 mM potassium phosphate buffer (pH 7), 7 mM (NH_4_)_2_SO_4_, 2 mM MgSO_4_, 10 μM FeSO_4_, 0.4% (wt/vol) glucose, 0.1% (wt/vol) casamino acids, 0.5% (wt/vol) agar. The bacterium was inoculated in LB and left to grow at 37 °C and 180 rpm overnight. The following morning, the bacterial suspension was used to perform a 1:100 dilution in LB broth. When the absorbance at 600 nm reached 1.0, 2 μL of broth culture was taken and seeded onto the center of the plates. The plates were incubated for 48 h at 37 °C. When required, spiramycin was used at a final concentration of 60 μg/mL.

### 4.7. Inoculum and Management of Galleria Mellonella Larvae

*Galleria mellonella* larvae were used as an in vivo infection model [78,95,96]. The inoculated larvae were selected to be the same size (approximately 2.5 cm in length). *P. aeruginosa* was grown as described above, and the bacterial culture was centrifuged (4000 rpm, 15 min, room temperature). The pellet was resuspended with an appropriate volume of a 10 mM MgSO_4_ solution to obtain a final absorbance at 600 nm = 1.0. Then, dilutions were then made to obtain a bacterial suspension containing 8 × 10^3^ CFU/mL. Next, 10 μL of this suspension were used to inject the larvae using a Hamilton syringe (250 μL) and an automatic dispenser. The injection was performed on the left pro-4th leg as described in the literature [97]. As a control, a spiramycin solution (60 μg/mL) was prepared by diluting the stock antibiotic solution into the MgSO_4_ resuspension solution.

Four sets of ten larvae each were used as control groups: (i) non-injected larvae (control larvae), (ii) larvae inoculated with resuspension solution alone (MOCK inoculum), (iii) larvae inoculated with a spiramycin solution without the bacteria (60 μg/mL) (spiramycin control), (iv) larvae infected with *P. aeruginosa*. A set of 10 larvae was injected with the bacterial resuspension with the addition of spiramycin (60 μg/mL). Larvae were incubated at 37 °C for 24 h [96]. After this time, the larvae were observed by counting the number of dead larvae (dark in color).

To measure the immune response of *G. mellonella* larvae to *P. aeruginosa* infection, the resuspension solution contained 8 × 10^5^ CFU/mL. Preparation and injection were performed as described above. The incubation was performed for 6 h at 37 °C. After this time, the larvae were frozen at −80 °C. RNA extraction from the larvae was performed as described below.

### 4.8. RNA Extraction from G. mellonella Larvae and RT-qPCR

Frozen larvae were harvested using sterile forceps, and then the larval segment corresponding to the 4th pair of pro-legs was removed using sterile scissors. The harvested tissue was immediately immersed into 950 μL of Trizol. The larvae pieces were finely minced using a sterile pipette and the liquid was collected after centrifugation at 12,000× *g* for 5 min at 4 °C. Subsequently, 200 μL of chloroform was added and the solution was mixed by inversion and incubated for 5 min at room temperature. After centrifugation at 12,000× *g* for 5 min at 4 °C, the aqueous phase was recovered, and total RNA was precipitated by adding an equal volume of isopropanol. The solution was incubated on ice for 10 min and then centrifugated at 12,000× *g* for 30 min at 4 °C. The supernatant was removed, and the pellet was dissolved with 500 μL of phenol solution pH 4.8, 250 μL of sterile distilled water, and 250 μL of chloroform. After centrifugation at 12,000× *g* for 5 min at 4 °C, the aqueous phase was recovered, and the RNA precipitated by adding 1/10 volume of 3 M sodium acetate (pH 7.0) and 2 volumes of cold ethanol, and incubated overnight at −20 °C. RNA was collected by centrifugation at 12,000× *g* for 30 min at 4 °C, and then washed with cold 70% ethanol, air dried, and resuspended in 100 μL of sterile water. RNA samples were treated with RNase-free Dnase (Promega, Madison, WI, USA) for 1 h at 37 °C and then at 75 °C for 5 min. The integrity and the concentration of the RNA samples were assessed by electrophoresis analysis on 1% agarose gel and UV-spectrophotometry (NanoDrop^®^, ND-1000 Spectrophotometer), respectively. For each experimental group, an equal RNA amount from three larvae was mixed at the final concentration of 1 µg/µL. Reverse transcription was carried out as previously reported [98]. The same RNA extraction protocol was used for *P. aeruginosa*. A volume of 1 mL was sampled at 24 h and 48 h during the growth of *P. aeruginosa* in the flasks, RNA was extracted, and a RT-qPCR was performed.

Quantitative gene expression analysis was carried out on CFX Connect™ Real-Time PCR Detection System (Bio-Rad Laboratories, Segrate, Italy), using SYBR^®^ Select Master Mix for CFX (Life Technologies, Carlsbad, CA, USA) and ubiquitin for normalization. The primers used for real-time PCR analysis are reported in Table 2 [78,80,99,100].

## Figures and Tables

**Figure 1 antibiotics-12-00499-f001:**
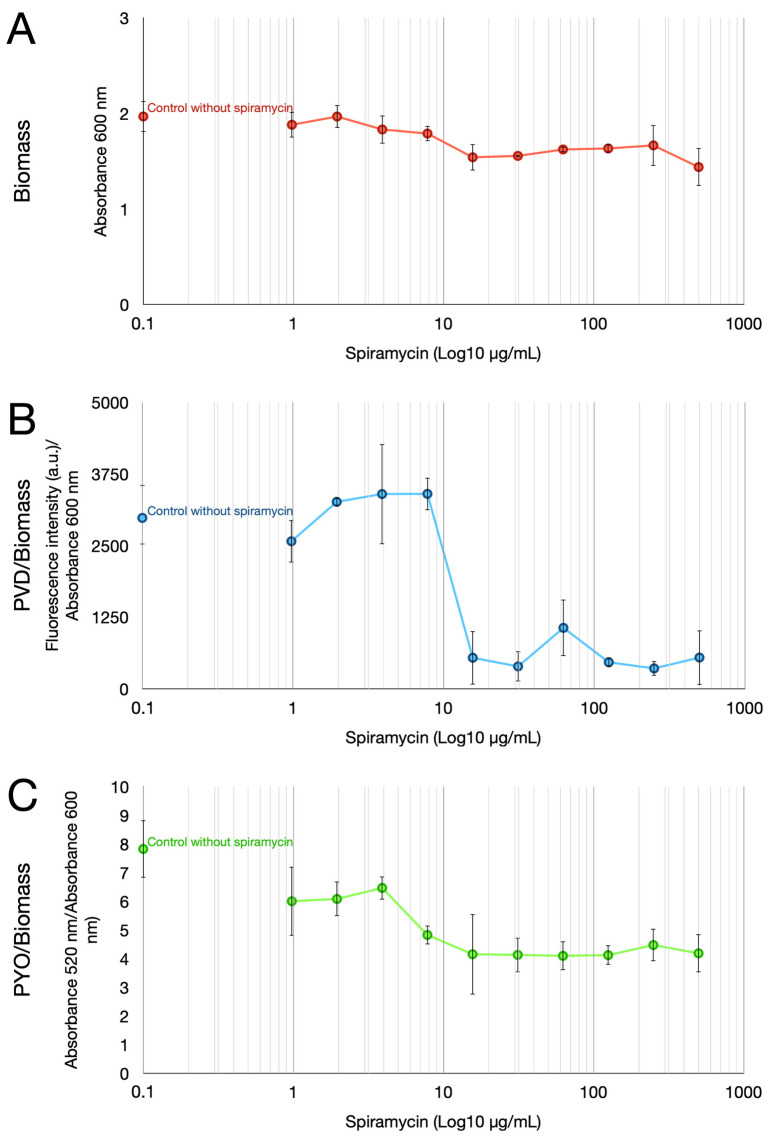
Biomass, PVD, and PYO production by *P. aeruginosa* GG-7R growing in LB for 24 h at 37 °C and 180 rpm. (**A**) Biomass was determined by absorbance at 600 nm measurement. (**B**) PVD in exhausted LB was assayed by fluorescence emission (excitation at 405 nm, emission at 450 nm). The value was normalized by the biomass (absorbance at 600 nm). (**C**) PYO in exhausted LB was assayed by optical absorbance at 520 nm. The value was normalized by the biomass (absorbance at 600 nm).

**Figure 2 antibiotics-12-00499-f002:**
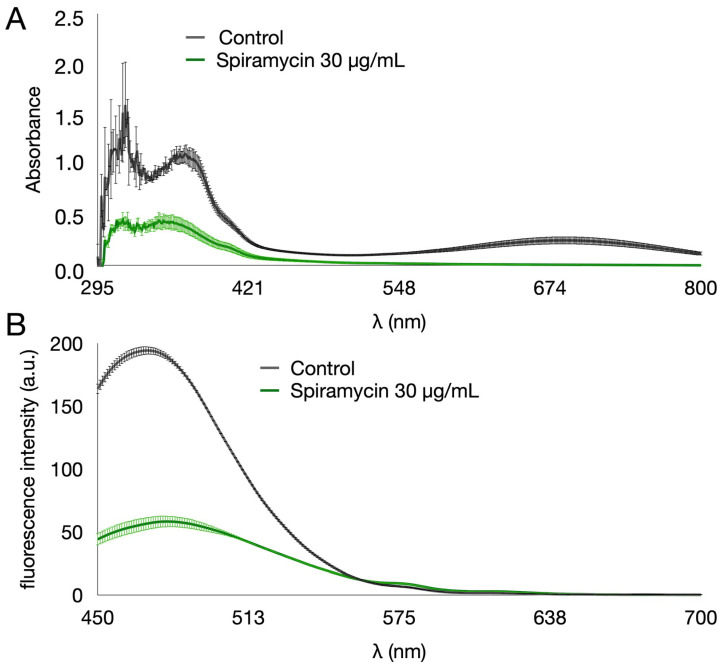
Absorption and fluorescence emission spectra of the exhausted LB broth of *P. aeruginosa* GG-7R grown either in the absence or in the presence of spiramycin (30 μg/mL). (**A**) Absorption spectra and (**B**) fluorescence emission spectra (excitation at 405 nm) of exhausted LB from *P. aeruginosa* cultures in multiwell plates. The samples were collected after the MIC experiment. The LB broth was diluted by 1 to 200 prior to measurement. The bacterium was grown at 37 °C and 180 rpm for 24 h.

**Figure 3 antibiotics-12-00499-f003:**
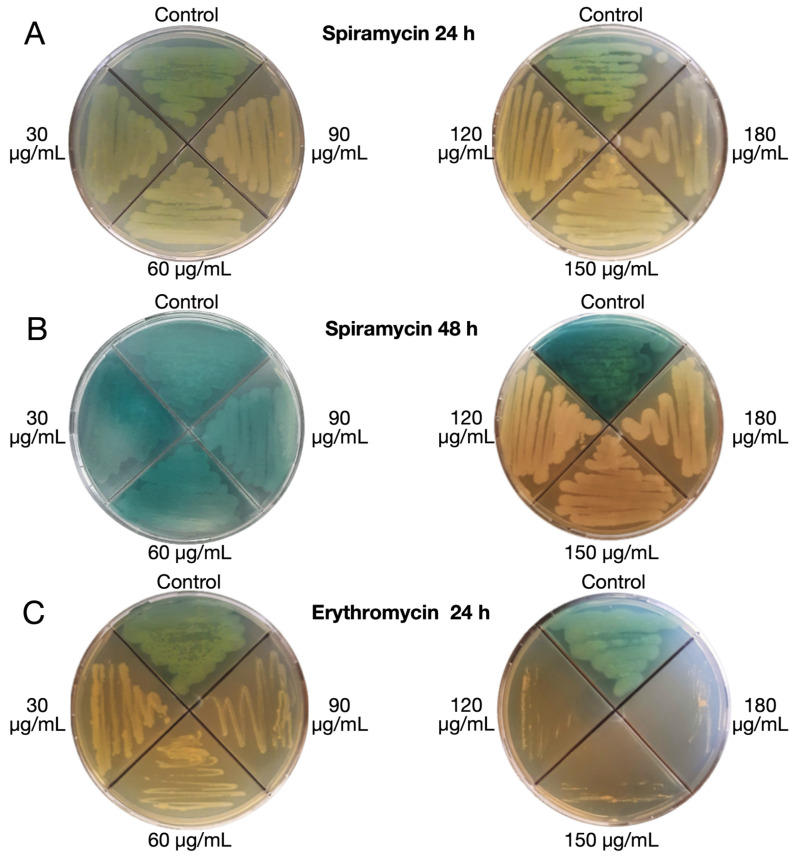
Phenotype of *P. aeruginosa* GG-7R growing on LB agar in the absence or presence of spiramycin or erythromycin. (**A**) Effects of spiramycin at 24 h of incubation (37 °C). (**B**) Effects of spiramycin at 48 h of incubation (37 °C). (**C**) Effect of erythromycin at 24 h of incubation (37 °C).

**Figure 4 antibiotics-12-00499-f004:**
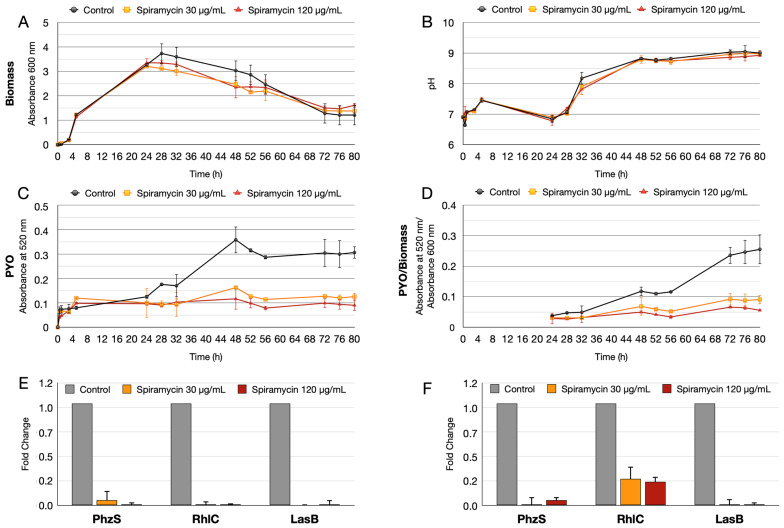
Growth of *P. aeruginosa* (37 °C and 180 rpm) with and without spiramycin (30 or 120 μg/mL) and RT-qPCR on *phzS*, *rhlC,* and *lasB* genes. (**A**) Estimation of biomass by turbidity (optical absorbance at 600 nm), (**B**) pH of bacterial cultures, (**C**) estimation of the amount of PYO (optical absorbance at 520 nm). (**D**) Estimation of the amount of PYO (optical absorbance at 520 nm) normalized by the biomass (absorbance at 600 nm). This panel shows data from 24 h onwards when the optical absorbance at 520 is >0.15 in the control sample. (**E**,**F**) Results of transcript level analysis (RT-qPCR) at 24 h (**E**) and 48 h (**F**) of the *phzS* (pyocyanin synthesis), *rhlC* (rhamnolipid synthesis), and *lasB* (elastase) genes.

**Figure 5 antibiotics-12-00499-f005:**
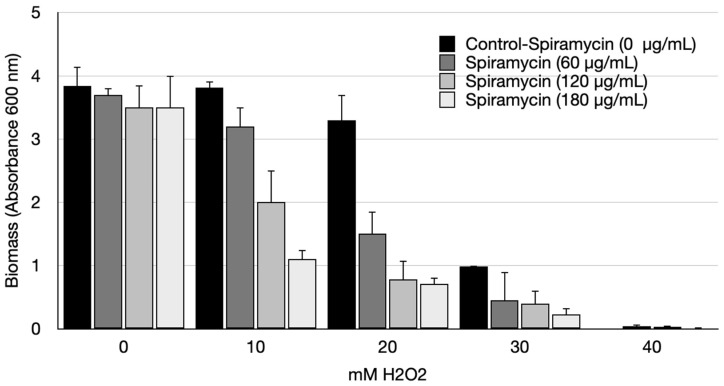
Growth of *P. aeruginosa* with hydrogen peroxide (H_2_O_2_) with or without spiramycin. The control value is indicated by a dotted line. Growth was carried out in LB at 37 °C and 180 rpm.

**Figure 6 antibiotics-12-00499-f006:**
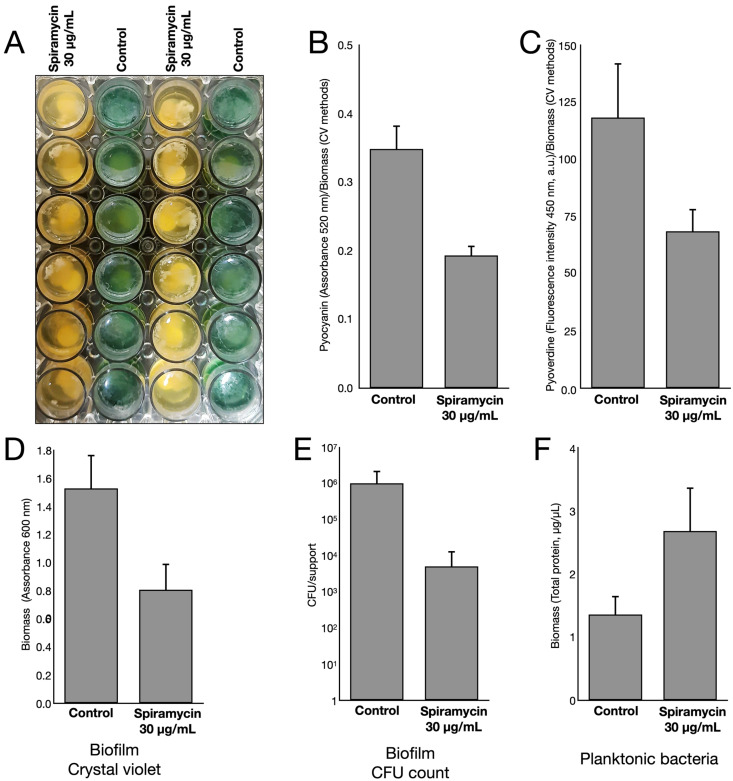
Effects of spiramycin on growth, biofilm, PYO, and PVD production by *P. aeruginosa* GG-7R growing on hydroxyapatite surfaces in multi-well plate filled with LB. (**A**) View of the plate after 72 h of incubation (37 °C, 150 rpm). (**B**) PYO quantification (optical absorbance at 520 nm) normalized by the biomass measured by the CV method. (**C**) PVD quantification (excitation at 405 nm, emission at 450 nm) normalized by the biomass measured by the CV method. (**D**, **E**) Estimation of the biomass using Crystal Violet method (**D**) or CFU counts (**E**). (**F**) Biomass of planktonic bacteria estimated by absorbance at 600 nm.

**Figure 7 antibiotics-12-00499-f007:**
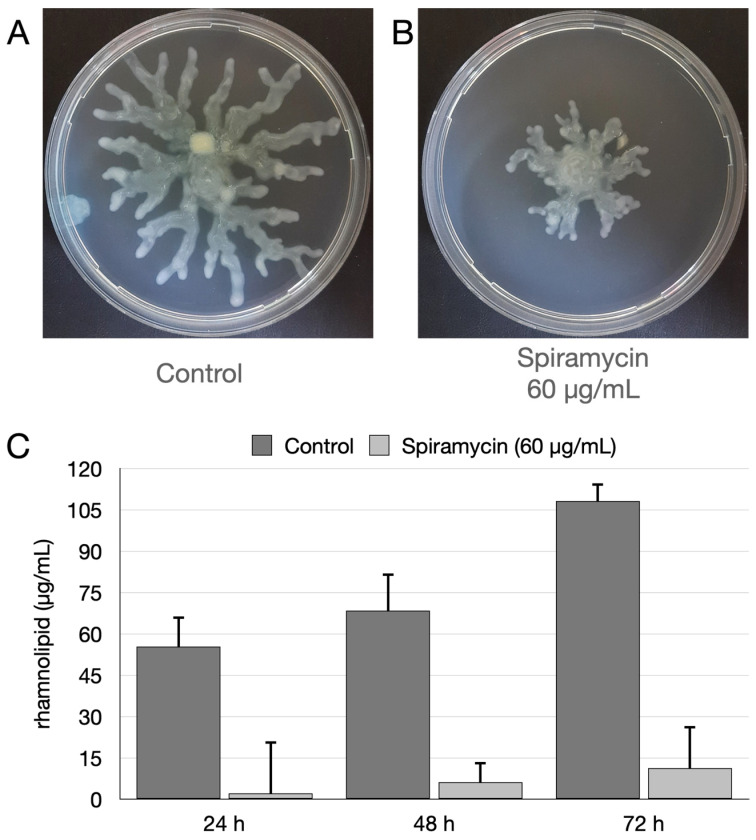
Effect of spiramycin on swarming motility and rhamnolipids production. The dishes were incubated at 37 °C for 48 h. (**A**,**B**) Growth and swarming motility of *P. aeruginosa* on BM2 solid medium in absence (**A**) or presence (**B**) of spiramycin (60 μg/mL). (**C**) Effects of spiramycin on rhamnolipid production.

**Figure 8 antibiotics-12-00499-f008:**
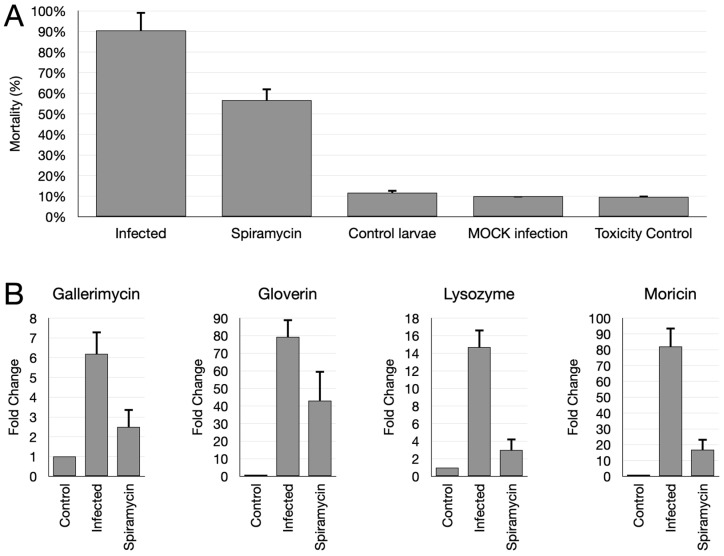
Effect of spiramycin on *G. mellonella* larvae infected with *P. aeruginosa*. (**A**) Mortality of *G. mellonella* larvae infected with *P. aeruginosa* treated or not with spiramycin 24 h after infection. Control larvae: non-injected larvae; mock infection: larvae injected with resuspension solution without bacteria; toxicity control: larvae injected with resuspension solution and spiramycin. (**B**) Transcript levels of genes encoding three antimicrobial peptides (gallerimycin, gloverin, and moricin) and lysozyme in *P. aeruginosa* infected larvae treated or not with spiramycin.

**Table 1 antibiotics-12-00499-t001:** MIC experiments: susceptibility of *P. aeruginosa* GG-7R at different antibiotics tested.

μg/mL	AMP	STR	RMP	ERY	SPM
500	-	-	-	-	+
250	+	-	-	-	+
125	+	-	-	+	+
62.5	+	-	-	+	+
31.3	+	+	-	+	+
15.6	+	+	+	+	+
7.8	+	+	+	+	+
3.9	+	+	+	+	+
2.0	+	+	+	+	+
1.0	+	+	+	+	+
0.5	+	+	+	+	+
0.2	+	+	+	+	+
Control	+	+	+	+	+

AMP = ampicillin; RPM = rifampicin STR = streptomycin; ERY = erythromycin; SPM = spiramycin. + = Growth; - =No growth.

**Table 2 antibiotics-12-00499-t002:** Primers used for real-time PCR analysis.

Sample	Primer Name	Sequence 5′-3′	Reference
*G. mellonella*	Gallerimycin f	GAAGTCTACAGAATCACACGA	[79]
*G. mellonella*	Gallerimycin r	ATCGAAGACATTGACATCCA	
*G. mellonella*	ubiquitin f ^1^	TCAATGCAAGTAGTCCGGTTC	[80]
*G. mellonella*	ubiquitin r ^1^	CCAGTCTGCTGCTGATAAACC	
*G. mellonella*	Gloverin f	GTGTTGAGCCCGTATGGGAA	[79]
*G. mellonella*	Gloverin r	CCGTGCATCTGCTTGCTAAC	
*G. mellonella*	Lysozyme f	GGACTGGTCCGAGCACTTAG	[79]
*G. mellonella*	Lysozyme r	CGCATTTAGAGGCAACCGTG	
*G. mellonella*	Moricin f	GCTGTACTCGCTGCACTGAT	[79]
*G. mellonella*	Moricin r	TGGCGATCATTGCCCTCTTT	
*P. aeruginosa*	*lasB* r	AACCGTGCGTTCTACCTGTT	[99]
*P. aeruginosa*	*lasB* f	CGGTCCAGTAGTAGCGGTTG	
*P. aeruginosa*	*rhlC* f	GCCATCCATCTCGACGGAC	[99]
*P. aeruginosa*	*rhlC* r	CGCAGGCTGTATTCGGTG	
*P. aeruginosa*	*phzS* f	CCGAAGGCAAGTCGCTGGTGA	[99]
*P. aeruginosa*	*phzS* r	GGTCCCAGTCGGCGAAGAACG	
*P. aeruginosa*	COM1 ^1^	CAGCAGCCGCGGTAATAC	[100]
*P. aeruginosa*	COM2 ^1^	CCGTCAATTCCTTTGAGTTT	

^1^ = gene used for normalization.

## Data Availability

All data are available in the text of the manuscript.

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
