# Peer review of "Spiramycin Disarms Pseudomonas aeruginosa without Inhibiting Growth"

_antibiotics, 2023, doi:10.3390/antibiotics12030499_

Round 1
Reviewer 1 Report
Matteo Calcagnile et al has shown in this investigation that, in the absence of any discernible impact on bacterial multiplication, spiramycin decreases the expression of virulence determinants in P. aeruginosa. An environmental strain of P. aeruginosa produced significantly less pyoverdine and pyocyanin, as well as swarming motility, rhamnolipid synthesis, and biofilm formation, in the presence of spiramycin, according to in vitro investigations. The study looks interesting for its further improvement I few minor suggestions.
Comments:
· In Table1 what indicates subscript 1 on all the antibiotic short form?
· Figure 2, I would suggest collecting OD of the exhausted LB and keep that data also in the graph for comparison, that LB itself has no influence in the reading.
· In Figure 2 legend the (A) and (B) should be written before the start of the figure description
· Figure 3 legend also the (A) and (B) description should be written properly
· Figure 4 legend write the full name of flask or remove the flask statement itself
· Figure 4 legend after (A) and (C) estimation is to be written as Estimation
· Figure 4 legend (C) from the agar plate assay indicates pigment formation in 48 hrs. but this assay indicates no pigment formation any explanation?
· Figure 4 legend (C) optical absorbance is different than OD if it is same then rewrite to OD. I would suggest keeping consistency in writing this makes the reader easy to understand
Author Response
Matteo Calcagnile et al has shown in this investigation that, in the absence of any discernible impact on bacterial multiplication, spiramycin decreases the expression of virulence determinants in P. aeruginosa. An environmental strain of P. aeruginosa produced significantly less pyoverdine and pyocyanin, as well as swarming motility, rhamnolipid synthesis, and biofilm formation, in the presence of spiramycin, according to in vitro investigations. The study looks interesting for its further improvement I few minor suggestions.
R. We thank the reviewer for the comments which allowed us to improve the manuscript.
Comments:
- In Table 1 what indicates subscript 1 on all the antibiotic short form?
R. The subscript is an indication for the reader as below the table is the full name of each antibiotic. As the subscript may cause confusion in the reader, we have chosen to remove it in the revised version of the manuscript.
- Figure 2, I would suggest collecting OD of the exhausted LB and keep that data also in the graph for comparison, that LB itself has no influence in the reading.
R. The LB medium spectrum alone was used as a blank to make the spectrophotometric measures in Figure 2. We have clarified this detail in Materials and Methods (ln.484-489) and in the caption of Fig. 2.
- In Figure 2 legend the (A) and (B) should be written before the start of the figure description
- Figure 3 legend also the (A) and (B) description should be written properly
- Figure 4 legend write the full name of flask or remove the flask statement itself
- Figure 4 legend after (A) and (C) estimation is to be written as Estimation
R. We have edited the legend of Fig. 2, Fig. 3 and Fig. 4 as requested by the reviewer.
- Figure 4 legend (C) from the agar plate assay indicates pigment formation in 48 hrs. but this assay indicates no pigment formation any explanation?
R. The experiment in figure 4 shows the growth and production of pyocyanin in liquid medium. The experiment of fig. 3, on the other hand, shows the growth of the bacterium on solid medium. In our experience, the production of pyocyanin is more efficient in LB agar than in LB broth.
- Figure 4 legend (C) optical absorbance is different than OD if it is same then rewrite to OD. I would suggest keeping consistency in writing this makes the reader easy to understand
R. We edited the text according to this comment.
Reviewer 2 Report
Please find attached my comments in the attached file.

Author Response
This manuscript by Calcagnile et al. presents both in vitro and in vivo experiments to demonstrate the inhibitory effect of a 16-membered macrolide, spiramycin, on the virulence of a global pathogen Pseudomonas aeruginosa. The authors first studied the absorption and emission spectra of the exhausted broth of P. aeruginosa in a multiwell-plate setup, finding that spiramycin inhibits the bacterial production of pyoverdine (PVD) and pyocyanin (PYO). Moreover, OD measurements showed that spiramycin doesn't affect much the bacterial growth. The authors confirmed these findings by looking at bacteria grown on agar plates or in flasks. Since PYO is a redox-active compound, the authors also assessed how spiramycin influences the growth of P. aeruginosa exposed to varying concentrations of H2O2. Indeed, the experimental results suggest that spiramycin sensitize P. aeruginosa to oxidative damage, presumably due to reduced production of redox-active compounds such as PYO. Besides PYO and PVD production, the authors further examined other key processes involved in the pathogenicity of P. aeruginosa and found that spiramycin also inhibited biofilm formation and swarming. Together, these results led the authors to test spiramycin's ability to repress virulence in vivo. Using Galleria mellonella larvae as the infection model, the authors showed that applying spiramycin to infected larvae greatly reduced their mortality as well as the expression of genes encoding antimicrobial peptides and lysozyme. Overall, this manuscript is reasonably well-constructed, and the results presented by this manuscript will be of interest to the field.
R. We thank the reviewer for his comments which allowed us to improve the manuscript. Following the reviewer's comments: i) we revised the figures (normalization for biomass), ii) we performed an RT-qPCR on three P. aeruginosa genes, iii) we re-performed the experiment with H2O2, and iv) we revised the text by adding some details.
Nevertheless, I have the following major suggestions/questions to the authors:
- For all the plots showing PYO and PVD production, are they normalized by OD or other measure of biomass? If so, please add a brief description in the main text or in the figure captions. If not, the authors need to remake these plots and show the normalized values to support their arguments. This comment applies to Fig.1B and C, Fig. 4 C, and Fig. 6B.
R. The values were initially reported as absolute. Following the reviewer's comment, we modified the figures. Fig. 1B and C and Fig. 6B were modified by replacing the reported panels. Fig. 2 was edited by adding a panel (D, normalized amount of pyocyanin per biomass). Details about normalization have been added to Materials and Methods (ln. 464-466, 487, and 526-527). Thanks to the reviewer for this helpful comment.
- I don't understand why there appears to be a discrepancy between Fig. 1C and Fig. 2A. I naively thought that the values in Fig. 1C would come from curves similar to those shown in Fig. 2A. However, the numbers for 30 µg/ml spiramycin and 520 nm don't match. The authors probably need to provide some explanation/clarification.
R. The spectra measured in Fig. 2 are from the spent LB broth taken from the experiment in Fig. 1. However, the broth was diluted 1 in 200 before being used to measure the continuous absorption and fluorescence spectrum. This allowed the instrument not to go above the detection limit (saturation). The point value at 520 nm in the presence of spiramycin is 0.022 which, when multiplied by the dilution of 200, gives 4.4. The dot shown in the graph in Figure 1 has a value of 4.13. Considering the error of the instruments used for dilution and considering that the measurement was made with two different instruments (Fig. 1 Cytation, Fig. 2 spectrophotometer) these two values can be considered equal. However, as the reviewer noted, this is not specified in the text. We have therefore modified both the legend in Fig. 2 and in the materials and methods section (ln., 484-489).
I also suggest adding the 0 µg/ml spiramycin control data to Fig. 1B and C. Fig. 2A shows that the 520 nm absorbance of the control and 30 µg/ml spiramycin experiments have at least several fold difference. However, the differences in absorbance at varying spiramycin concentrations are much less significant in Fig. 1C. I wonder if adding the control in 1C will make the difference more notable.
R. We have added point 0 µg/ml spiramycin to the plots in Figure 1.
- In Fig. 5, I don’t understand what the "control" value stands for. I thought there should be a different control value for every H2O2 concentration tested. I also don't understand why the control value is higher than the three bars at 0 mM H2O2. I thought the point is that spiramycin doesn't affect much the bacterial growth (in the absence of H2O2). In any case, I think adding the 0 µg/mL spiramycin results for each H2O2 concentration would be informative. My last question is: with 60 and 120 µg/mL spiramycin, why do the bacteria grow better at 10 mM H2O2 than at 0 mM H2O2? I wonder if the authors know or could speculate why this is the case.
R. Thanks to the reviewer for this comment. We re-performed this experiment, and we added the growth values obtained with spiramycin and without hydrogen. Also, we modified the graph by adding the value of the control (spiramycn 0 µg/mL).
- The experiments on the PYO and PVD production presented in this manuscript are great, but I also wonder whether the authors have tested/plan to test the effect of spiramycin on the PYO and PVD production via qPCR of genes involved in those processes. This will provide more direct support for the conclusion than the spectroscopic measurements.
R. Thanks to the reviewer for this comment. Following this indication, we performed RT-qPCR on three genes: phzS (pyocyanin synthesis), rhlC (di-rhamnolipids synthesis), and lasB (elastases). The results confirmed that spiramycin markedly reduced the transcript levels of these genes.
- How do the results shown in this manuscript generalize to other P. aeruginosa strains? Have the authors tested other more commonly used lab strains such as PA01 or PA14? I think adding one experimental example or even some discussion along this line will greatly generalize the current findings.
R. The effect of spiramycin on other strains of P. aeruginosa will certainly be tested in the future. For the time being, we have followed the reviewer's comment and added some line to the discussion (ln. 413-419).
Minor points:
- There're several places where a sentence is too long. One example is the final sentence of the first paragraph of the Introduction section (lines 38-43: Drug repositioning is particularly valuable... the number of approved antibiotics). Another example can be found in the last sentence of the second paragraph (lines 47-54: In this context, ..., EKAPE pathogen ...). I suggest breaking long sentences into several short ones for clarity.
R. Thanks to the reviewer for this comment. We have modified some sentences making them shorter (ln. 38-44, 49-55, 72-77, 88-93, 104-107, 134-138, 160-164, 383-387, 391-395).
- Lines 66-67 (...led to a survival rate of 80% compared with 20% in controls.): I suggest replacing "20% in controls" with "a 20% survival rate in the control" for clarity.
R. We edited the manuscript as requested.
- Line 67 (These findings led to speculate that.. .): This part is grammatically incorrect. It should be either "These findings led [someone] to speculate that..." or "These findings led to the speculation that ..."
R. We edited the manuscript as requested.
- Line 93 (..., which differs from erythromycin for a larger macrolactone ring and ...): Isuggest replacing "for'' with "by having".
R. We edited the manuscript as requested.
- Table 1: The header of the first column should be "µg/mL" as is elsewhere. Also, it might be worthwhile explaining what "+" and "-" stand for in the footnote.
R. We edited the table as requested.
- Line 126 and line 170 (even if it did slightly reduce the final biomass values ...): I think "even though" is more appropriate here than "even if."
R. We edited the manuscript as requested.
- Line 149 (at all tested concentrations (30, 60, 90, 120, 150 and 150 µg/mL)): The last number should be "180" instead of "150".
R. We edited the manuscript as requested.
- Line 151 (The ability of spiramycin (120, 150 and 150 µg/mL)...): Same problem as above."
R. We edited the manuscript as requested.
- Line 170 (The results confirmed that spiramycin did not inhibit the bacterial, ...): Missing a word "growth" before the comma."
R. We edited the manuscript as requested.
- In Fig. 3, I wonder what would happen at even later time points with spiramycin. This will clarify whether spiramycin simply delays the production of PYO (and long delay with higher concentration of spiramycin) or it reduces the production of PYO whatsoever. I understand that the colony will enter the stationary phase later, which might complicate the interpretation of the experiment, but it might still be worth looking at this.
R. We observed the plates shown in the figure for about a week. After 48 h (72 h, 96 h and so on) no changes were detected. However, we have not reported this observation because after 48 h the cultures are at confluence, as noted by the reviewer, and therefore it is not possible to know whether the effect is due only to the presence of spiramycin or determined by senescence in the bacterial culture. In accordance with this comment, we have added these details in the text (ln. 157-160)
- Line 194-196 (... , the biomass at 24 h decreased significantly as a function of spiramycin concentration, compared with corresponding samples untreated with spiramycin.): I would replacing "compared with corresponding samples untreated with spiramycin" with "and was the highest when the cultures were untreated with spiramycin."
R. We edited the manuscript as requested.
- Lines 232-233 (Furthermore, rhamnolipid production has been associated with the motility patterns of aeruginosa.): I suggest directly saying "rhamnolipid production promotes twitching/swarmig motility". This way, the results in Fig. 7C would be more naturally expected."
R. We edited the manuscript as requested.
- Line 233 (Thus, rhamnolipids were quantifies the exhausted broths): Typo. "quantifies" needs to be replaced by "quantified".
R. We edited the manuscript as requested.
- In Fig. 8, I'm guessing that "control larvae" indicates the non-injected larvae and "toxicity control" indicates the larvae injected with spiramycin alone. The authors need to explain what these two phrases mean in the figure caption or in the main text.
R. We edited the caption of Fig. 8 according to this comment.
- Line 295 (..., as inhibitors...): I assume as stands for "quorum sensing" here. The authors need to define the acronym earlier in the text.
R. We defined the acronym QS in the introduction. In the discussion, we replaced the words quorum sensing with the acronym QS.
- Lines 323-324 (the growth of P. aeruginosa was not affected even in the presence of very high concentration of spiramycin, ...): To help clarify the result, Isuggest rewriting "in the presence of very high concentration of spiramycin" as "with the highest concentration of spiramycin tested (500 ug/mL)".
R. We edited the manuscript as requested.
- Lines 340-342 (..., inhibition of expression of the GacA-dependent small RNAs RsmY and RsmZ of the which modulate upstream of the quorum-sensing machinery.): I suggest adding an "and" in front of "inhibition". Also, this part seems to be missing a word I several words. Maybe "RsmY and RsmZ, both of which modulate..."?
R. We edited the manuscript as requested.
- Lines 411-412 (The amount of PYO in the exhausted broths was measured using the absorbance measurement (520 nm).): I suggest adding a reference on why 520 nm absorbance is used for quantifying PYO production.
R. We edited the manuscript as requested.
Reviewer 3 Report
Calcagnile and colleagues wrote a paper entitled "Spiramycin Disarms Pseudomonas aeruginosa Without Inhibiting Growth" in which they showed how spiramycin, although not effective on Pseudomonas aeruginosa, inhibits the expression of virulence determinants in P. aeruginosa. In vitro experiments showed that the production of pyoverdine and pyocyanin in P. aeruginosa strains was significantly reduced in the presence of spiramycin, as well as biofilm formation. They also determined the immunomodulatory effect of spiramycin. Accordingly, they speculated that spiramycin could contribute to the treatment of infections caused by P. aeruginosa in combination with other antibiotics.
The topic of the paper is very interesting, the experimental methods are appropriate, and the results are clearly presented.
The literature used to write this paper is very old, only 15% of the literature is within the last 5 years. Given the topicality of the topic and the existence of numerous recent papers, I do not understand why this is so? I believe that newer literature should be added to the paper.
References are not written according to MDPI guidelines, please correct
It is necessary to correct:
Line 96: (MLS) should be written (MLSB)
Line 386-403: Part of the paper written in Materials and Methods under the title "4.1. Strain, media, growth conditions and general procedure" is not correctly written. That is, it is not clear whether the authors used a commercial broth, in which case they must indicate the manufacturer. It is not clear how they added spiramycin and erythromycin. Similarly, the manufacturer of the antibiotics used must be indicated.
Line 672: The year 2026 is incorrectly entered.
Author Response
Calcagnile and colleagues wrote a paper entitled "Spiramycin Disarms Pseudomonas aeruginosa Without Inhibiting Growth" in which they showed how spiramycin, although not effective on Pseudomonas aeruginosa, inhibits the expression of virulence determinants in P. aeruginosa. In vitro experiments showed that the production of pyoverdine and pyocyanin in P. aeruginosa strains was significantly reduced in the presence of spiramycin, as well as biofilm formation. They also determined the immunomodulatory effect of spiramycin. Accordingly, they speculated that spiramycin could contribute to the treatment of infections caused by P. aeruginosa in combination with other antibiotics.
The topic of the paper is very interesting, the experimental methods are appropriate, and the results are clearly presented.
R. We thank the reviewer for the comments which allowed us to improve the manuscript.
The literature used to write this paper is very old, only 15% of the literature is within the last 5 years. Given the topicality of the topic and the existence of numerous recent papers, I do not understand why this is so? I believe that newer literature should be added to the paper.
R. We edited the manuscript by adding more recent references (n° 25, 42-49, 65, 82-91, 99)
References are not written according to MDPI guidelines, please correct
R. We revised the references according to MDPI. If requested, other changes will be made during the correction of the proof.
It is necessary to correct:
Line 96: (MLS) should be written (MLSB)
R. The error has been corrected.
Line 386-403: Part of the paper written in Materials and Methods under the title "4.1. Strain, media, growth conditions and general procedure" is not correctly written. That is, it is not clear whether the authors used a commercial broth, in which case they must indicate the manufacturer. It is not clear how they added spiramycin and erythromycin. Similarly, the manufacturer of the antibiotics used must be indicated.
R. We thank the reviewer for this comment. We have improved the text by adding manufacturers of antibiotics and components used to prepare the media. In addition, we have better described how the media were prepared (ln. 440-446).
Line 672: The year 2026 is incorrectly entered.
R. The error has been corrected.
Round 2
Reviewer 2 Report
I am pleased by the modifications that the authors made to the manuscript, which addressed the concerns I raised previously. My recommendation is that the manuscript should be accepted.